# Multi-Level Context Pyramid Network for Visual Sentiment Analysis

**DOI:** 10.3390/s21062136

**Published:** 2021-03-18

**Authors:** Haochun Ou, Chunmei Qing, Xiangmin Xu, Jianxiu Jin

**Affiliations:** School of Electronic and Information Engineering, South China University of Technology, Guangzhou 510640, China; hchuno@163.com (H.O.); xmxu@scut.edu.cn (X.X.); jxjin@scut.edu.cn (J.J.)

**Keywords:** sentiment analysis, emotion, context, MCPNet, MACM

## Abstract

Sharing our feelings through content with images and short videos is one main way of expression on social networks. Visual content can affect people’s emotions, which makes the task of analyzing the sentimental information of visual content more and more concerned. Most of the current methods focus on how to improve the local emotional representations to get better performance of sentiment analysis and ignore the problem of how to perceive objects of different scales and different emotional intensity in complex scenes. In this paper, based on the alterable scale and multi-level local regional emotional affinity analysis under the global perspective, we propose a multi-level context pyramid network (MCPNet) for visual sentiment analysis by combining local and global representations to improve the classification performance. Firstly, Resnet101 is employed as backbone to obtain multi-level emotional representation representing different degrees of semantic information and detailed information. Next, the multi-scale adaptive context modules (MACM) are proposed to learn the sentiment correlation degree of different regions for different scale in the image, and to extract the multi-scale context features for each level deep representation. Finally, different levels of context features are combined to obtain the multi-cue sentimental feature for image sentiment classification. Extensive experimental results on seven commonly used visual sentiment datasets illustrate that our method outperforms the state-of-the-art methods, especially the accuracy on the FI dataset exceeds 90%.

## 1. Introduction

Studies have shown that image sentiment affect visual perception [1]. Compared with the non-emotional stimulus content in the image, the affective content attracts the attention of the viewer more strongly, and the viewer has a more detailed understanding of the affective stimulus content [2]. Therefore, the purpose of visual sentiment analysis is to understand the emotional impact of visual materials on viewers [3], which plays an important role in opinion mining, user behavior prediction, emotional image retrieval, game scene modeling and other aspects.

Initially inspired by psychological and artistic principles, researchers studied color, texture and other hand-crafted features at the image level for visual sentiment analysis. Like many visual tasks, gradually, Convolutional Neural Networks (CNN) replaces the hand-crafted features because it can automatically learn the deeper representation of images, and the actual research also proves that the CNN-based method is obviously superior to the hand-crafted features.

However, visual sentiment analysis is inherently more challenging than traditional visual tasks (such as object classification, scene recognition, etc.) [4], mainly because it involves more complex semantic abstraction and subjectivity [5]. Existing works illustrate that the viewer’s emotional changes are often caused by certain areas of the image [6] because of the selective attention [7], so many of recent studies have proposed different methods on how to use local area representation to improve classification accuracy [4,8]. However, most of the methods extract local features from the final feature map of backbone, so that the obtained features are single-level and can perceive objects of extremely limited scale. Based on the co-existence of sentiment evoked and the objects in local region [9,10], visual sentiment analysis needs to solve the following two basic problems more than many traditional visual tasks in image content understanding [11,12]:Perception of different scale objects. The size and location of objects in images from social networks are diverse, which means that we need to have multi-scale analysis capabilities in the model. As shown in Figure 1, the scale of people in the images (a) to (c) is from small to large. Thus, a single-scale method object perception can only capture objects at a limited scale, while losing object information at other scales.Different levels of emotional representation. Different objects can evoke dissimilar degrees of sentiment. As shown in Figure 1e–h. Some simple objects contain less semantic information, such as the flower (e), and the street lamp (f). The emotional stimulation they express is weak, and their emotional information can be described by the low-level features. However, the complex semantic information will give us stronger emotional stimulation such as humans and human-related objects. Human’s non-verbal communication such as facial expression (g), body language and posture (h), have a strong ability to express emotions [13]. These complex objects need more abstract high-level semantic features to describe their emotional information.

In addition, the viewer’s attention is affected by different levels of features, such as low-level attributes (for example, intensity, color) and high-level semantic information [14]. Figure 1d is also a person holding an umbrella, but due to the different colors of the umbrella, our focus is different from Figure 1b,c. Therefore, the methods only use the final high-level semantic features of the model are insufficient.

In this paper, a multi-level context pyramid network (MCPNet) composed of multi-scale adaptive context modules (MACM) is proposed to deal with the above two problems at the same time. Multi-scale, adaptive context modules combined with different levels of features are presented to capture more different cues to improve model performance. Moreover, compared with the current local area analysis method that does not consider the relationship between different areas, our method can adaptively learn the relationship between different regions in the image to obtain the areas related to sentiment.

In summary, the contributions of this paper can be highlighted as follows:Adaptive context framework is introduced for the first time in the image sentiment analysis task. This method can learn the correlation degree of different regions in the image by combining different scale representations, which is helpful to improve the ability of the model to understand complex scenarios.The multi-scale attributes in the proposed MACM module are alterable. Compared with many existing multi-scale methods that can only capture objects of fixed limited scales, our method can combine different scales to capture objects of different positions and sizes in the image.The proposed MPCNet adopts cross-layer and multi-layer feature fusion strategies to enhance the ability of the model to perceive semantic objects at different levels.The experiment proves the advancement of our method, and the visualization results show that our method can effectively identify the small semantic objects related to emotional expression in complex scenarios.

## 2. Related Work

In this section, we will review the CNN method with additional information and the region-based CNN method. The difference between the two is that the CNN model of the former is based on the existing excellent image classification model, while the model of the latter is designed for visual sentiment analysis. Then we will introduce the context-based approaches that are closely related to our method.

### 2.1. CNN with Additional Information

The hand-crafted features have limitations, while the features extracted by excellent CNN show better results than the hand-crafted features on different datasets [9,15]. On this basis, researchers began to introduce more additional information to help CNN model improve performance. Borth et al. [16] proposed a form of multiple adjective noun pairs to describe image content, called ANPs; Chen et al. [17] further proposed a classification CNN called DeepSentiBank based on visual sentiment concepts. Li et al. [18] further combined the text information in ANPs and calculated the emotional information of the text value in ANPs in the form of weighted sum. Yuan et al. [19] proposed an algorithm called Sentribute with 102 mid-level attributes that is readily comprehensible and can be used for higher-level sentiment analysis. Yang et al. [20] introduced the probability distribution constraint of sentiment in the loss calculation of single label classification task. Kim et al. [21], Ali et al. [22] used the excellent model of target recognition and scene classification competition to generate objects and scene category labels as high-level semantic features for image sentiment classification.

The additional information of the above methods mainly comes from the annotation information. The cost of introducing additional annotation information is expensive, and thus this approach is not suitable for large-scale datasets. Because visual sentiment analysis is more difficult than general visual tasks, it is necessary for us to design corresponding models or modules specifically for visual sentiment analysis tasks.

### 2.2. Region-Based CNN

Studies have shown that the expression of human emotion is related to the region of local concern in the image, and different regions may have different effects on the expected expression [1,23]. Recently, local region-based image sentiment analysis methods have achieved encouraging performance improvements on many image sentiment datasets. Zhou et al. [24] showed the potential of CNN in local learning. Peng et al. [23] proposed that sentiment can be induced by specific regions, and proposed EmotionROI dataset. Fan et al. [1] proposed an affective priority effect through comparative experiments. Yang et al. [25] proposed WSCNet, which weights the final output features of the network by cross spatial pooling module and integrates local image features into classification. Song et al. [8] integrated the concept of visual attention into visual sentiment classification, using multi-layer CNN to model the attention distribution of the whole image which constraining by saliency detection, and located the region with the largest amount of information to improve the classification performance. On the basis of NASNet, Yadav et al. applied the residual attention module to learn the important areas related to emotion in the image [26]. Wu et al. suggested to use the object detection module to determine if use a local module [27]. Yang et al. [28] proposed a method to finding related regions using a ready-made method to get object proposals as local emotional information and used VGG to learn global information. According to Rao (c) [4], most of the current methods only consider the use of the final level features, ignoring the contribution of different levels of features to visual sentiment expression, which limits their model performance. Therefore, based on Faster R-CNN [29], they further improved the model performance by combining different levels features and the global information. Their experimental results show that they are currently the best results.

### 2.3. Context-Based CNN

Most of current methods try to obtain the emotional local area of the image, and treat these local areas independently, without considering whether these local areas related to emotion are connected. This is not conducive to the analysis of complex scenarios. Contextual information can help the model understand complex scenarios [12] and are widely used in scene parsing and semantic segmentation tasks. On the basis of existing work, APCNet [12] further summarizes the role of the multi-scale, adaptive and global information guidance context features in understanding complex scenes. A single scale can only capture objects of single scale, and there is information loss on other scales. Therefore, it is necessary to use multi-scale in most of visual tasks. The attribute of adaptive makes a pixel in the context feature not only associate with the pixel nearby, but also infer the relationship between the pixel and other regions in the global context, which can capture the long-distance dependences between the pixel and the regions. Therefore, the adaptive attributes can help us to obtain the relationship between different regions.

Inspired by APCNet, we introduce multi-scale adaptive context features into visual sentiment analysis tasks for the first time. The multi-level features can combine the details and the semantic information from different level features. The low-level features can describe the simple-looking objects, and the high-level features are more suitable for describing complex-looking objects.

## 3. Methodology

In this section, we will introduce the proposed multi-level context pyramid network (MCPNet) in detail. The framework is illustrated in Figure 2, which includes multi-scale adaptive context modules (MACM) from different levels. The proposed MACM can learn the degree of association of different regions in the image at different scales for corresponding level. In order to make better use of different levels of features, we use the cross-layer and multi-layer feature fusion strategies to combine different levels of features from MACM to obtain multi-cue sentimental feature for classification.

### 3.1. Proposed Multi-Level Context Pyramid Network

The architecture of proposed MCPNet is illustrated in Figure 2. The ResNet101 is utilized as backbone, because it is a commonly used and effective model in visual tasks, and ResNet101 has enough convolutional layers to help us extract multi-level features. As shown in Figure 2, the ResNet101 network can be divided into four parts: c2, c3, c4, and c5, representing different levels of features. Considering the amount of computation, we only take the features of the output of c3, c4, and c5. Among features at different levels, low-level features have more detailed information, but weaker semantics representation and more noise. High-level features can represent more complex semantic objects but lacking in detail. For c3, c4, and c5, different level features are sent to the corresponding MACM to obtain the corresponding cross-layer multi-scale context features called $O^{3},O^{4},O^{5}$ as the local emotional representation. For different levels of contextual features, a cross-layer feature fusion strategy is adopted to increase the relationship between adjacent levels of features. Then multi-level contextual features are combined with global emotional representation to form the multi-cue emotional feature E for sentiment classification. In Section 4.6, we will further verify the effectiveness of the multi-cue emotional feature E through visualization.

### 3.2. Multi-Scale Adaptive Context Module

MACM is the core module of the context pyramid network, which aims to calculate the contextual features of each location by using the local region sentiment affinity coefficient of different scales under global guidance. The process of MACM is illustrated in Figure 3. For the input I, $X^{l}$ is the feature map from the layer l of backbone, and $X_{i}^{l}$ is the representation at the position of feature i in $X^{l}$. The value of l is c3, c4, or c5. In order to get the context features adaptively with different scales on $X^{l}$, $Z_{i}^{l}$ is introduced to represent the multi-scale context feature of $X_{i}^{l}$:(1)$$z_{i}^{l}=F_{context}^{l}\left(X^{l},i\right)$$

MACM consists of two branches. The following shows how MACM works through one layer of feature $X^{l}.$

(1) Sub-regions Branch: This branch is to learn the local sub-region representations of the input feature map $X^{l}$ under different scale divisions. For each scale $S_{k}$, the feature map $X^{l}$ is divided into $S_{k}\times S_{k}$ sub-regions $Y^{s_{k}}$:(2)$$Y^{s_{k}}\;=\;\left[Y_{1}^{s_{k}},Y_{2}^{s_{k}},\ldots ,Y_{s_{k}\times s_{k}}^{s_{k}}\right],k=1,2..n$$

For each sub region $Y_{j}^{s_{k}}$, the feature will be extracted by averaging pooling and 1×1 convolution. In implementation, the feature of $Y^{s_{k}}\in R^{s_{k}\times s_{k}\times 512}$ under the scale $S_{k}$ is extracted through an adaptive average pooling with a 1×1  convolution. Then, $Y^{s_{k}}$ will be reshaped to $Y^{s_{k}}\in R^{s_{k}^{2}\times 512}$ to match the shape of the affinity coefficient $\alpha^{s_{k}}$ of the corresponding scale calculated by the other branch.

(2) Region Sentiment Affinity Coefficient Branch: The purpose of this branch is to learn the affinity coefficient weights between sub-regions at the same scales under the guidance of global information. The region sentiment affinity coefficient $\alpha_{i,j}^{s_{k}}$ is introduced to represent the degree of association of the sub-region $Y_{j}^{s_{k}}$ with the sentiment of the estimated $X_{i}^{l}$. In order to realize $\alpha_{i,j}^{s_{k}}$ local feature association property, $X^{l}$ firstly will be sent to the resize operation to get $M^{l}\in R^{14\times 14\times 512}$. The resize operation is realized by controlling the parameters of convolution kernel, as showing in Figure 4. There are 1×1 and 3×3 convolution kernels in this operation. After 1×1 convolution, the length and width of the feature map remain unchanged, and the number of channels is 512; after the convolution of 3×3, the channel becomes 512, and the length and width become half of the original.

After that, by using global average pooling for $M^{l}$, the global information characterization $g(M^{l}$) is obtained. Then, $M^{l}$ and $g(M^{l}$) are multiplied and calculate the local affinity vector for each local location i under the global perspective:(3)$$\alpha_{i,j}^{s_{k}}\;=\;f_{s_{k}}\left(M_{i}^{l},g\left(M^{l}\right),j\right)$$

In implementation, $f_{s_{k}}$ is achieved by a 1×1 convolution and sigmoid activation function. For each location i, the affinity vector corresponding to each scale is $s_{k}\times s_{k}$, which corresponds to the number of sub-regions in this scale. So, it has a total of h×w affinity vectors, each of which has a length of $s_{k}^{2}$ and reshaped it to the size of $hw\times s_{k}^{2}$.

(3) Computing multi-scale context feature Z: For each scale $s_{k}$, the region sentiment affinity coefficient $\alpha_{i,j}^{s_{k}}$ indicates the degree of relevance of the sub-region $Y_{j}^{s_{k}}$ to the sentiment of the estimated $X_{i}^{l}$. The single-scale adaptive context vector $W_{i}^{s_{k}}$ of position i for scale $s_{k}$ can be calculated as
(4)$$W_{i}^{s_{k}}=\;\overset{s_{k}\times s_{k}}{\underset{j=1}{\sum }}\alpha_{i,j}^{s_{k}}Y_{j}^{s_{k}}$$

For the context feature $W^{s_{1}},W^{s_{2}},\ldots ,W^{s_{n}}$ obtained by different scale calculation sequence, context features of different scales will be combined, and then do batch normalization (BN) after a 1×1 convolution. Finally, the multi-scale context feature $Z_{i}^{l}$ for $X_{i}^{l}$ will be obtained. The details are as follows:(5)$$Z_{i}^{l}=\;f\left(\left[W_{i}^{s_{1}},W_{i}^{s_{2}},\ldots ,W_{i}^{s_{n}}\right]\right)$$

$Z^{l}$ represents the multi-scale adaptive context feature of the l layer, and through cross-layer feature fusion, the multi-scale emotional representation of l layer is obtained.

### 3.3. Cross-Layer and Multi-Layer Feature Fusion Strategies

In order to enhance the connection between features of different depths, here two fusion strategies are adopted to balance semantic information and detailed information.

(1) Cross-layer features fusion: Among the features of different levels, the semantic and detailed information of the features of two adjacent levels is the closest. The idea of fusion of features of two adjacent layers has been reflected in Feature Pyramid Networks (FPN). However, unlike FPN, the shape of the context feature $O^{l}$ output by each layer of MACM is 14 × 14 × 512, it can’t use an up-sampling method for feature fusion between adjacent layers. To reduce the noise from low-level features, we have adopted the same shape of $M^{l+1}$ and $O^{l}$ for feature fusion. There are different numbers of 3×3 convolutions using to transform the feature maps of different layers to obtain $M\in R^{14\times 14\times 512}$. In each layer, multiple scales are used to capture semantic information of different sizes and positions. In the network, we adopt three scales of 1, 2, and 4. In particular, when s =1, $\alpha_{i,j}^{s_{k}}$ represents the global weight of each location, which is the feature learning under the global perspective. Then, the context features of all scales are fused to obtain the multi-scale context feature $Z^{l}$ of this layer. Then for different layers, adding context information $Z^{l}$ obtained by the MACM module of this layer and the $M^{l+1}$ obtained from the previous layer to get the output feature $O^{l}$ of this layer. Details as follow:(6)$$\{\begin{array}{c} O_{i}^{l}=Z_{i}^{l}\times M_{i}^{l}\;,\;l=c5\\ \;\;\;O_{i}^{l}=Z_{i}^{l}\times M_{i}^{l+1}\;,\;l\neq c5 \end{array}$$

After cross-layer feature fusion, the shapes of the obtained O are the same, which is conducive to further multi-level feature fusion.

(2) Multi-layer features fusion: To increase the global and local connections, we also combine the global information of the input image and the context features $O^{3},O^{4},O^{5}$ obtained from layers c3, c4, and c5. Specifically, the last output $X^{5}\in R^{14\times 14\times 2048}$ of backbone and $O^{3},O^{4},O^{5}$ are concatenated together to get the multi-cue sentimental feature E:(7)$$E=\left[X_{global},O\right]=\left[X^{5},O^{3},O^{4},O^{5}\right]$$

Then, after global average pooling, it enters the full connection layer for classification.

## 4. Experiment

### 4.1. Dataset

The proposed method is evaluated on seven widely used datasets of different sizes, including IAPSsubset [30], ArtPhoto [31], Abstract Paintings [31], Twitter I [32], Twitter II [16], EmotionROI [23] and FI [15]. Some examples as shown in Figure 5 and the details are illustrated in Table 1.

Small-scale dataset: IAPSsubset comes from International Affective Picture System [33] and has 395 images in eight emotional categories. Different from other datasets, ArtPhoto is a dataset composed of 806 art photos, and Abstract Paintings consists of 228 abstract pictures of colors and textures. Twitter I and Twitter II were collected from social network Twitter and labeled by AMT workers with 1,269 and 603 images. We set up experiments similar to [28,32] on all three subsets of Twitter I. EmotionROI is developed from Emotion6 [34], and the image comes from the Flickr website. Compared to Emotion6, EmotionROI adds 15 emotion-related annotation boxes to each image marked by participants and believes that the more repeated annotation boxes on a pixel point, the greater the contribution of the point to emotional expression.Large-scale dataset: FI is currently the most commonly used large-scale visual sentiment dataset, which is collected through social network using emotional categories as search keywords. 225 participants from AMT were employed to label resulting in 23,308 images.

In visual sentiment analysis tasks, different labeling methods are used, and the number of categories in the dataset is different. At present, there is no uniformity in the number of dataset categories in visual sentiment analysis tasks. Due to the influence of subjectivity, the dataset of visual sentiment analysis task requires expensive manual annotation [35]. The six used affective datasets contain less than two thousand images, except the FI dataset (see Table 1), which are far from the required number for training robust deep networks. Therefore, in this paper, we focus on binary emotion prediction (positive and negative) and convert the emotional labels of Mikel [30] and Ekman [36] into the original binary affective tags, which are compared with existing advanced methods in the above datasets.

### 4.2. Implementation Details

The proposed MCPNet uses ResNet-101 pre-trained on ImageNet [37] as the backbone. Before training, random horizontal flipping and clipping random 448 × 448 patches are used as data augmentation to reduce over fitting. We use the SGD optimizer and the momentum is 0.9. The learning rate is set to 0.001 and decreased 10 times every 7 epochs of 100 epochs. FI dataset was randomly divided into 80% for training, 5% for validation and 15% for testing. Other datasets were randomly divided into 80% training and 20% testing [16], except for datasets with specifying training/test partition [23,31]. Further proves the effectiveness of our method through 5-fold cross validation experiment in the above 7 datasets. All our experiments were carried out on one NVIDIA GPU and completed all codes and experiments through Pytorch.

### 4.3. Baseline

In the following, we will evaluate our methods with the state-of-the-art algorithms of image sentiment classification, including the hand-crafted features and deep learning methods.

#### 4.3.1. Hand-Crafted Features

GCH [38]: the global view of image with 64-bin color histogram features.

LCH [38]: the local view of image with 64-bin color histogram features.

PAEF [39]: This research is one of the early works in visual sentiment analysis that focuses on more complex features than the low features. It contains low-level and middle-level features inspired by artistic principles.

Rao(a) [40]: It is an early exploration of analyzing local areas related to sentiment. Dividing a picture into different blocks through image segmentation, called multi-scale blocks, and SIFT-based bag-of-visual features contain local and global information extracted from the image blocks.

SentiBank [16]: Proposing a 1200-dimensional middle-level feature called adjective noun pairs (ANPs) to describe the relationship between image content and sentiment. This work is an important work that explored the correspondence between semantic information and emotion in the early stage.

#### 4.3.2. Features Based on CNN

AlexNet [41]/ VGG-16 [42]/ ResNet101 [37]: pre-trained on ImageNet, and then fine-tuned the FC layer on the sentiment dataset.

DeepSentiBank [17]: This is a work to improve SentiBank and propose a CNN called DeepSentiBank. Compared with SentiBank, this work proposes 2089-dimensional ANPs, which significantly improves labeling accuracy and retrieval performance.

PCNN [32]: a progressive training framework based on VGGNet. They use a large amount of weakly supervised data to let the model learn some common visual features to reduce the difficulty of training the visual emotion dataset.

Rao(b) [9]: multi-level deep features from the framework based on AlexNet with side branch. This is an important exploration of the emotional analysis method using CNN for multi-level features.

Zhu [43]: a CNN framework containing a bidirectional RNN module with multi-task losses for visual emotion recognition.

AR [28]: This work puts forward a new concept called Affective Regions in the research of exploring local regions and sentiment evoked and use the ready-made object detection technology as local information combined with VGG model for analysis.

RA-DLNet [26]: The residual attention module is applied in NASNET to focus on the local areas of the image which are related to emotion.

GM_EI_ &LRM_SI_ [27]: The work found through research that not all images in the dataset contain salient objects. Therefore, this work believes that visual sentiment analysis should not only focus on local features. The work respectively proposed global modules and local modules, and determine whether to use a local module through the object detection module.

Rao(c) [4]: This work argues that most of the current researches on visual emotion analysis only focus on high level features and do not pay attention to the effects of different level features on visual emotion tasks. They use the multi-level framework of object detection algorithms to obtain the local representations that trigger emotions, are combined with the output features of the last layer of backbone to classify image emotions as global features.

### 4.4. Experimental Validation

#### 4.4.1. Choice of Scale s

In the proposed MCPNet, as shown in Figure 3, different s represents different scales used by the model, and we give the corresponding classification performance on the FI dataset. For feature maps at different levels, we unify them into one size. Higher-level feature has larger perception fields. It is not that the larger the scale, the better the performance. The performance of different s by taking it from 1 to 6 is shown in Figure 6. When s = 1, 2, 4, the model achieves the top three performance respectively. Therefore, our experimental setup adopts s = 1, 2, 4. The results of subsequent experiments with different datasets also show that our selection of s is equally effective on other datasets. Of course, for different datasets, we can better adapt to different task by choosing different s combinations. However, here we want to use a more general combination of s.

#### 4.4.2. Effective of Different Level Features

Low-level and high-level features have different effects on sentiment classification How to combine the two is a consideration for many visual tasks. To provide more cues to the task of visual sentiment analysis, previous studies have shown the effectiveness of using multi-level features in experimental results [4,9]. Here, we further explore the role of different levels of features. As can be seen from Table 2, $O^{5}$ represents the context of high-level features, with the largest impact on the model. When $O^{5}$ is removed and the model performance drops by almost 2%. $O^{3}$ denotes the context of the underlying features and has the smallest impact. One interesting finding, however, is that removing $O^{5}$ or $O^{4}$ has little difference. We think this is due to more than half of the backbone convolution being performed at the c4 level. Because we just remove $O^{5}$ or $O^{4}$, not c4 or c5. Therefore, the model accuracy of using only $O^{5}$ or $O^{4}$ is still 89.011 and 88.668, which illustrates that both of them have a significant contribution to model performance. The accuracy was declined almost 10 percentage points by only using $O^{3}$. This indicates that the model performance can be improved in varying degrees by using different level of features.

Here it also showed the effect of removing global information and the performance of the model decreased by 0.52% on FI. This indicates that it is necessary to combine global information with the context features. We also performed comparative experiments on different levels of context feature fusion strategies. When no cross-layer fusion strategy was used, the model decreased by 0.3%. Further performance changes are shown in Figure 7 and Figure 8. They show that the contributions of $O^{4}$ and $O^{5}$ to the model performance are close, but the effect of $O^{3}$ is much lower than that of $O^{4}$ and $O^{5}$. In this regard, it can be seen that on the large dataset such as FI, it is not enough to only use the low-level features.

### 4.5. Comparisons with State-of-the-Art Methods

This section shows the comparison between the proposed MCPNet and the state-of-the-art method on the seven datasets. The experimental settings are mainly referred to by Rao(c). To more fully reflect the performance of our methods, all of methods are used five-fold cross-validation to get the final classification score.

As shown in Table 3, the results illustrate that the performance of the CNN methods far exceeds the hand-crafted methods. Hand-crafted features only perform well in some small datasets such as Abstract, ArtPhoto, and EmotionROI. Mainly because the Abstract is composed of abstract paintings and ArtPhoto is composed of art photos. They are greatly affected by the low-level features like color and texture. EmotionROI actively eliminates images that evoke sentiment with some high-level semantic such as facial expressions and text during the collection process. Therefore, some hand-crafted methods perform better than some deep learning methods on these three datasets. For example, the Rao(a) method is nearly 1.2% higher than ResNet01 on Abstract. However, the hand-crafted method is far inferior to the CNN method on datasets with more high-level semantic information such as IAPSsubset and the large dataset FI. With fewer layers, AlexNet is still nearly 6% higher than Rao(a) in both FI and IAPSsubset. Region-based methods such as AR and Rao(c) have significantly higher performance than global image-based methods. Our method mines sub-region relations and their long-distance dependencies in images, which have multi-scale characteristics.

Rao(c)’s method [3] obtains local areas of different scales through faster R-CNN based on FPN which requires a tedious pre-training process that the entire model needs to be pre-trained on the COCO [44] and EmotionROI datasets to obtain object perception capabilities. However, our model does not require other additional cumbersome pre-training processes. In our method, the feature map of the input MACM module is divided into different sub-regions, and the relationship between different regions is learned to obtain different scale perception abilities. Compared with methods such as Rao(c), our method is 2.8% higher on FI and 2.16% higher on EmotionROI. Our performance is also better than Rao (c) on other small datasets as shown in Table 3. To our knowledge, it is the first time that the accuracy of binary classification task over 90% on FI dataset. FI dataset is a large dataset, the others are small datasets and they rely on the pre-trained model on FI to continue training. Therefore, the proposed MCPNet has the better improvement effect on the FI dataset then other dataset. Figure 9 shows the accuracy of our method on the 5-fold cross-validation strategy, which is trained five times on the FI and EMotionROI. The dashed line represents the accuracy of Rao(c). Thus, our method is better than the existing methods in binary classification performance, which demonstrates the effectiveness of adaptive context features in visual sentiment analysis tasks.

Table 4 shows the classification results on Twitter I and Twitter II. It can be seen that our method also advances. Although the result of our method is lower than GM_EI_ & LRM_SI_ 0.4% on Twitter I_4, and lower than RA-DLNet 0.01% on Twitter II dataset. However, from the perspective of these two data sets as a whole, the proposed MACPNet is still more advanced than these two methods.

### 4.6. Visualization

Same as other visual classification problems, for visual sentiment analysis tasks, an important question is whether the proposed model can recognize the affective areas or objects in the image. This is important for us to evaluate and understand the model. In this section, we use the Class Activation Mapping method [45] to further evaluate our model by visualizing the multi-cue sentiment feature $E\in R^{\left(2048+512\times 3\right)\times 14\times 14}$. Further evaluations were made on the EmtionROI and EMod datasets. EMod is a dataset specifically designed for the study of visual saliency and image sentiment analysis. It contains 1019 images coming from IAPS and Google’s search engine, and each image has eye tracking data annotated by 16 subjects. In order to evaluate these two datasets uniformly and to further test the robustness of our method, we abandon the conventional practice of training the model on the corresponding dataset before visualizing it.

But do visualization experiments on EmtionROI and EMod separately after training the models on large datasets FI. The visualization results are given in the form of heat map. A comparison of the visualization results using a single scale with the scale combinations used in our model is given.

As shown in Figure 10 and Figure 11, there are obvious differences in the objects or regions captured with different scale. However, when the objects’ size of the same class varies greatly within the same dataset like person in Figure 10 and Figure 11, the single scale is difficult to fully handle. Even different scales can correctly perceive the same object, the areas of interest at different scales are biased. As shown in Figure 10 e,f, both scales can correctly focus on squirrels and humans. However, different scales focus on different positions of the squirrel’s head and human body. It may even be that in the case of Figure 11f, the correct object is not well attended by a single scale, requiring a comprehensive multi-scale analysis. Therefore, multi-scale attributes are necessary in visual affective analysis. More results on the EmotioROI and EMod are given in Figure 12. From Figure 12, our visualization results are surprising. From some difficult situations such as people from far away (d1, h4), small flies on paper cups (c1), to faces with different sizes and appearances (a4, c4, d4, i4, j1, j4), our model all illustrates excellent object perception. However, it is worth noting that our model is not trained on these two datasets.

In Figure 13, we also give some representative examples of failure. For simple natural scenes such as (a) and buildings in (b), the eye tracking collection of many taggers is not uniform and does not focus on a certain area well. This is because there is no obvious object to attract the viewer in the image. The affective signal is from the overall content of the image, while the eye tracking data of the viewer come from their own habits. It is also very difficult to the model to locate similar objects with the same type and small differences, such as (c). Human-related or human-specific semantic information in images can attract us more effectively [1], such as plain text, as shown in (d) in Figure 13. However, for sentiment analysis of images, the understanding of plain text in Figure 13d is extremely difficult.

## 5. Conclusions

In this paper, we propose two attributes for visual sentiment analysis model: multi-scale perception and different levels of emotional representation. Furthermore, a novel multi-level context pyramid network composed of multi-scale adaptive context modules is proposed to learn the sentiment correlation degree of different regions for different scale in the image. The adaptive context framework is introduced for the first time in the image sentiment analysis task, which is helpful to improve the ability of the model to understand complex scenarios. The multi-scale attribute of the proposed MACM module is independent of the model structure and can combine different scales according to different data sources to mine the semantic information of images, which has good selectivity and scalability and can capture objects of different positions and sizes in the image. Two different level features fusion strategies are analyzed in the model to make better use of different levels of emotional representation to enhance the ability of the model to perceive semantic objects. The experimental results illustrate the performance advantage of our method on different datasets. Furthermore, the visualization results also show that the proposed MCPNet can effectively perceive some extreme situations, such as extremely small objects in the image.

## Figures and Tables

**Figure 1 sensors-21-02136-f001:**
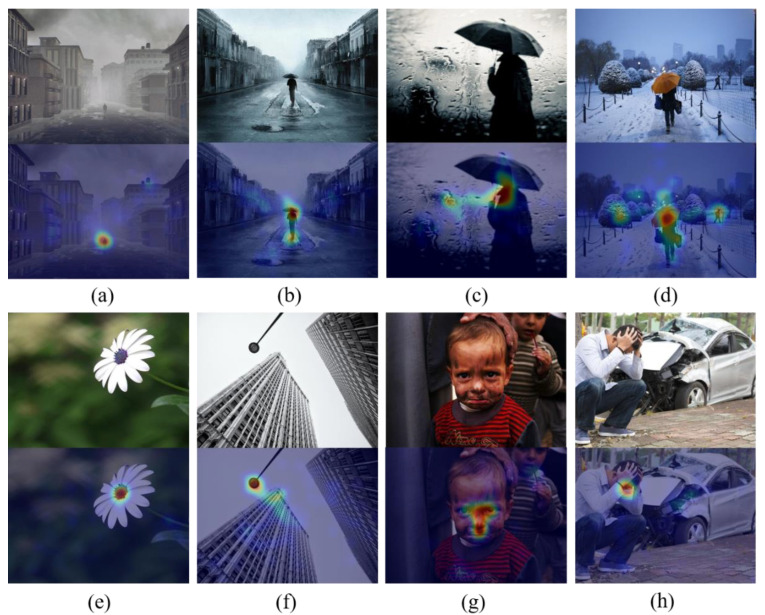
Images are from the emotional attention dataset EMOd [1]. In the eight columns of images from (**a**–**h**), the above is the original image, and the below is the eye-tracking focus of the image. Images (**a**–**d**) represent the backs of people, but their scales are significantly different, representing the different size of objects in the images. Images (**e**–**h**) represent different levels of semantic content from simple to complex.

**Figure 2 sensors-21-02136-f002:**
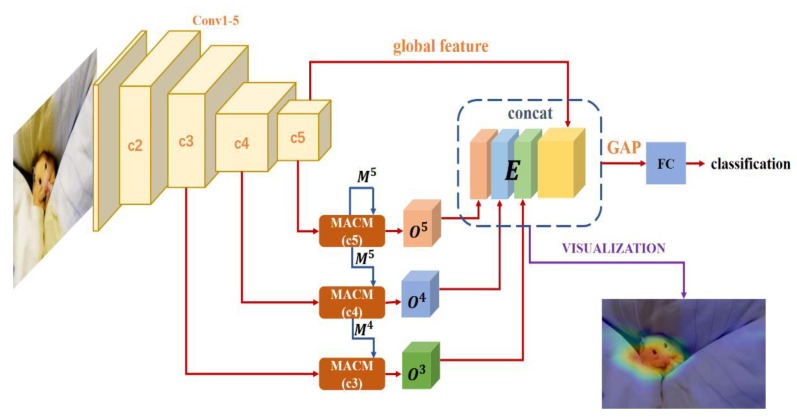
The framework of multi-level context pyramid network (MCPNet). Firstly, the input image is sent to backbone ResNet101 to get the different level features. The features of c3, c4, and c5 are put into MACM to obtain multi-scale context features, and then $O^{3},O^{4},O^{5}$ and c5 layer features will be combined to obtain multi-cue emotional feature E for visual emotion analysis. In this figure, concat is a channel-wise concat, GAP means global average pooling, FC means fully connected layer.

**Figure 3 sensors-21-02136-f003:**
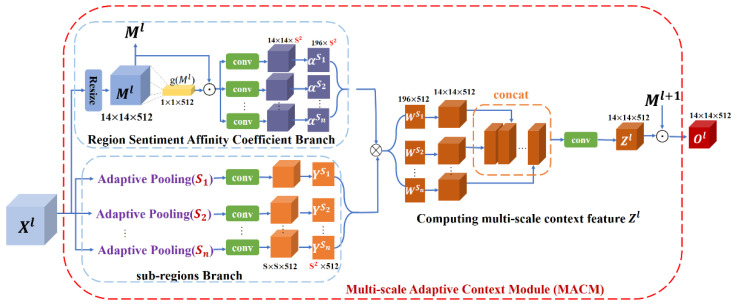
The process of MACM module. For the input feature graph $X^{l}$, there are two branches. One is sub-regions branch. Another one is region sentiment affinity coefficient branch. Finally, the multi-scale context features are fused with the $M^{l+1}$ of the upper layer. In this figure, concat is a channel-wise concat.

**Figure 4 sensors-21-02136-f004:**
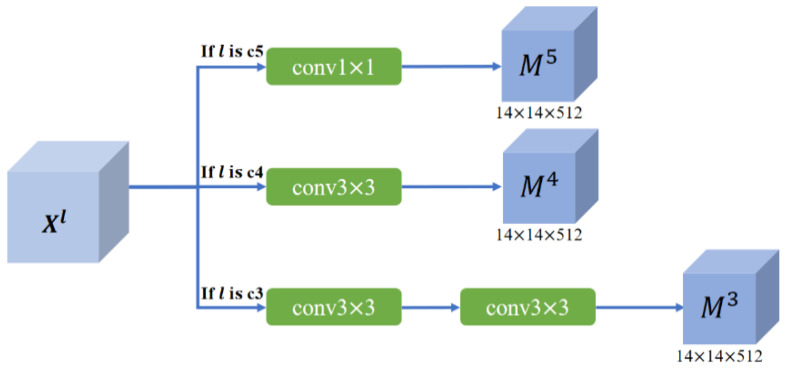
The resize operation for different level layer.

**Figure 5 sensors-21-02136-f005:**
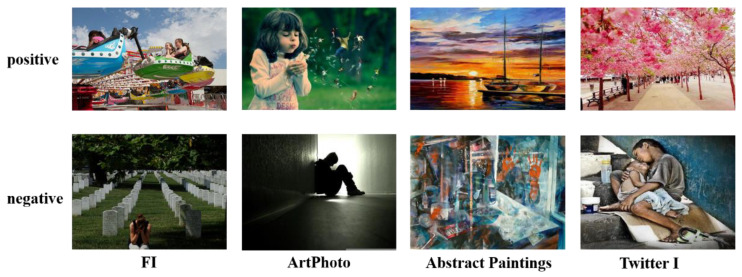
Some examples from the datasets we used.

**Figure 6 sensors-21-02136-f006:**
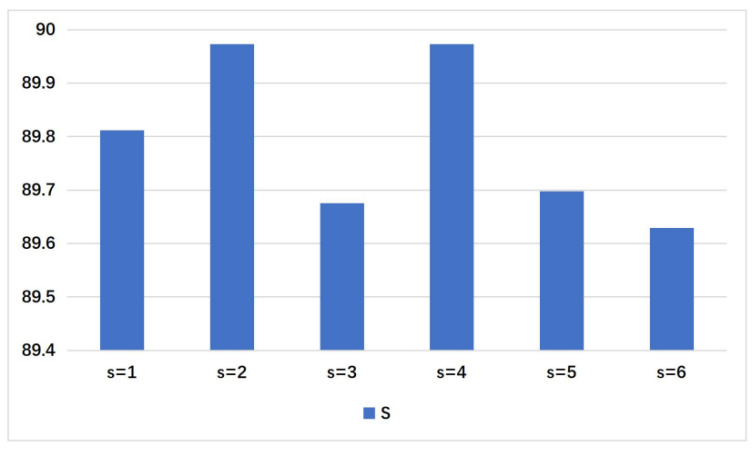
The verification of the impact of different scale s on FI dataset.

**Figure 7 sensors-21-02136-f007:**
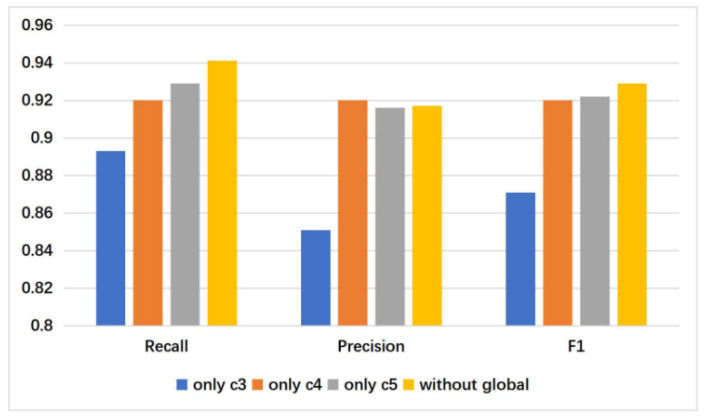
The performance with different levels of context features and global features removed on FI dataset.

**Figure 8 sensors-21-02136-f008:**
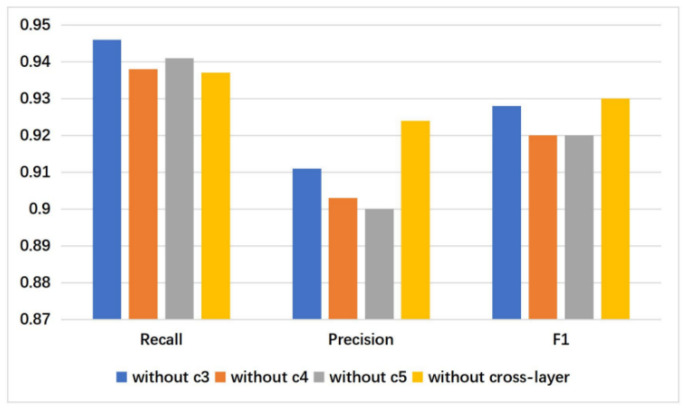
The performance without different-layer context features and cross-layer feature fusion on FI dataset.

**Figure 9 sensors-21-02136-f009:**
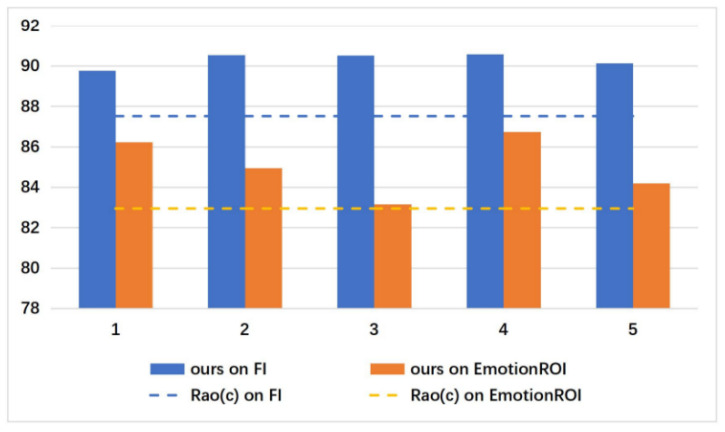
The results of our method on 5-fold cross validation on FI and EMotionROI. The final results are compared with the Rao (c)’s method, which has the highest accuracy presently.

**Figure 10 sensors-21-02136-f010:**
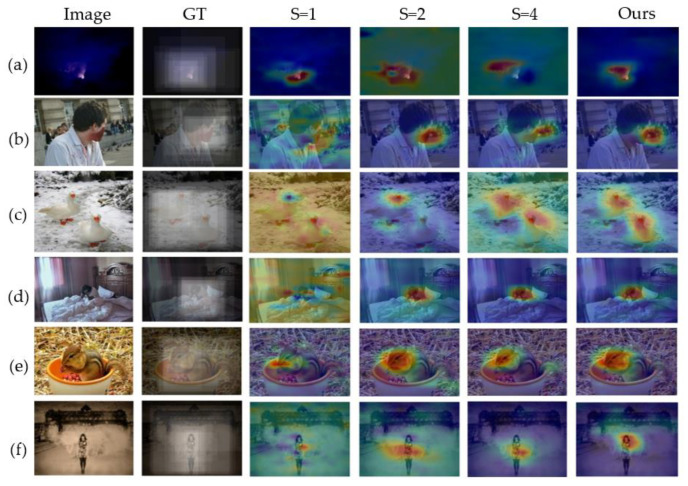
Visualization of the proposed model on EmotionROI. Here we compare with different scale of S = 1, S = 2, S = 4. Our method uses S = [1,2,4]. GT is the ground truth. As shown in (**b**–**d**), S = 2 and S = 1 are better than S = 4. However, in figure (**a**) the case of S = 1 is better than the case of S = 2 and S = 4. (**e**,**f**) reflect the different effects of different scales on the same object. These examples show that multi-scale models have more stable object perception capabilities than single-scale models.

**Figure 11 sensors-21-02136-f011:**
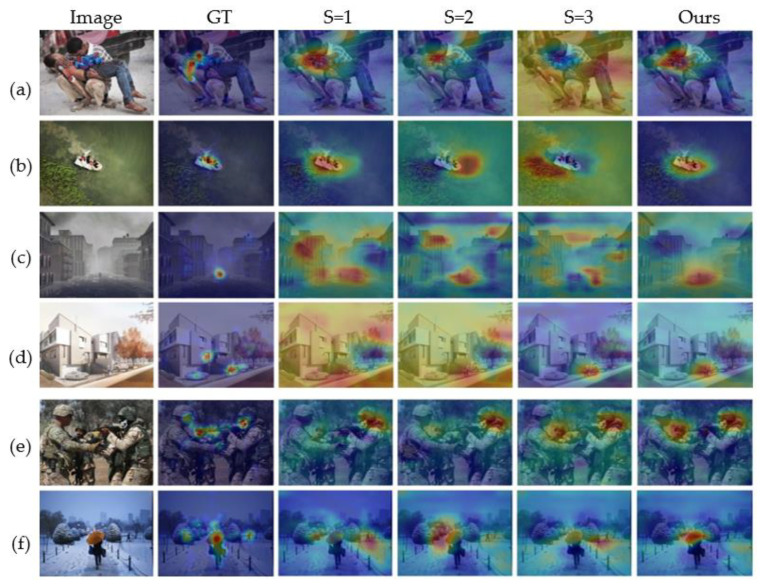
Visualization of the proposed model on EMod. Here we compare with different scale of S = 1, S = 2, S = 4. Our method uses S = [1,2,4]. GT is the ground truth. As shown in (**a**–**e**), sometimes single-scale or multi-scale can effectively perceive objects related to sentiment, but in complex situations such as (**f**), multi-scale is more effective than single-scale.

**Figure 12 sensors-21-02136-f012:**
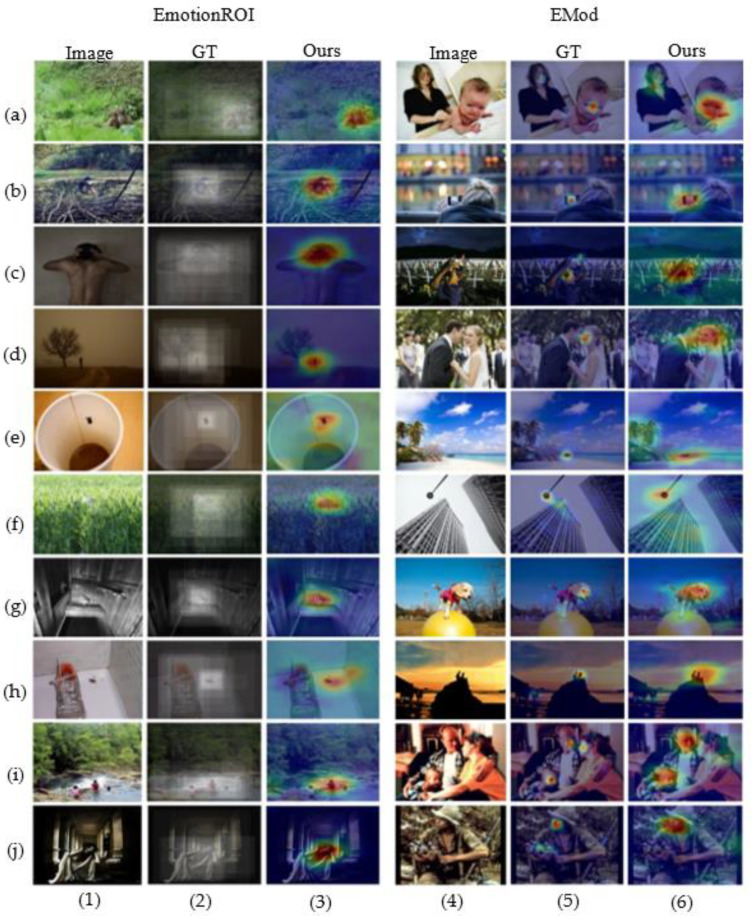
More visualization comparisons on EmotionROI and EMod. Here the complexity of appearance change varies with different objects. For the small-sized objects, such as (f1) birds in grass, (g1) masks on chairs, (b1, i1, h4) people, their appearance and surroundings are quite different. Since our model is not trained on these two datasets, the comparisons on these two datasets are fairly. GT is the ground truth.

**Figure 13 sensors-21-02136-f013:**
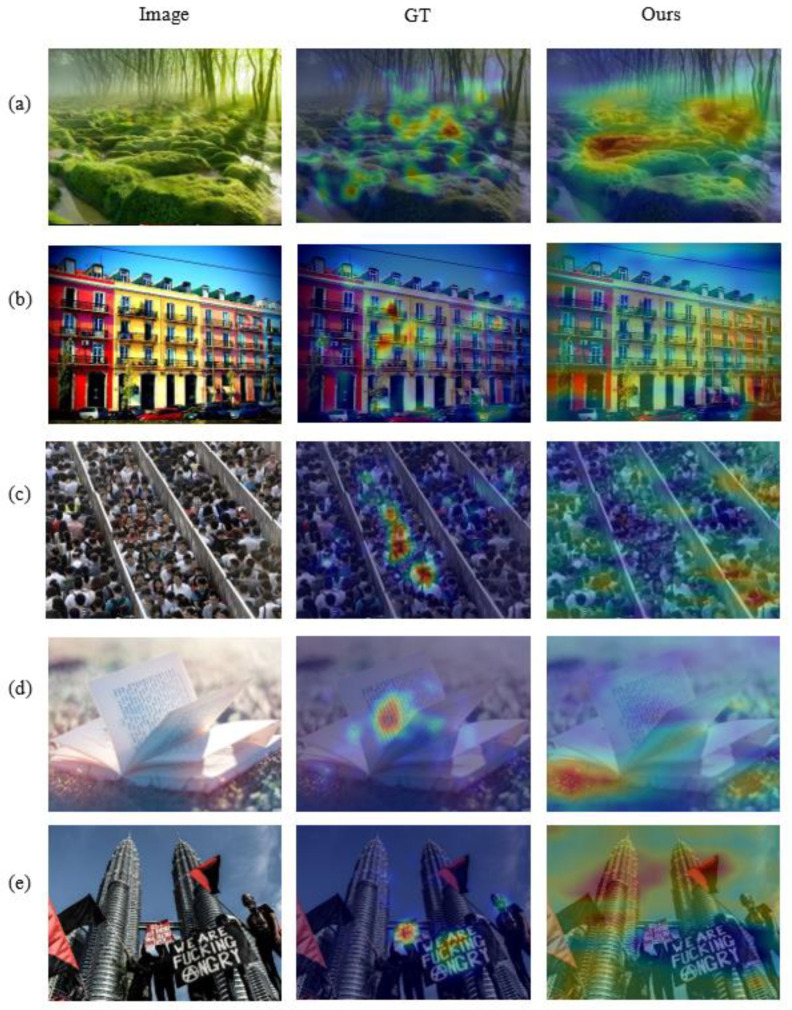
Some typical examples of failures on the EMod dataset. GT is the ground truth. For example, there are no prominent semantic objects in (**a**,**b**), and (**c**) exemplifies that when there are a large number of similar objects in the scene (people in this example), the visualization of the model in these examples is terrible. Human-specific factors are also difficult for the model. For example, the viewer can understand the text information in (**d**,**e**), but the model does not work.

**Table 1 sensors-21-02136-t001:** Seven commonly used visual sentiment analysis datasets. Most datasets are from social networks. Except for FI dataset, the others are relatively small datasets, because of the subjective and labor-intensive labeling process.

Dataset	Positive	Negative	Sum	Type
IAPSsubset [30]	209	186	395	natural
Abstract [31]	139	89	228	abstract
ArtPhoto [31]	378	428	806	artistic
Twitter I [32]	769	500	1269	social
Twitter II [16]	463	133	596	social
EmotionROI [23]	660	1320	1980	social
FI [15]	16430	6878	23308	social

**Table 2 sensors-21-02136-t002:** Comparison of different configurations of our model on FI dataset, including “only” and ”without”. For example, “only” of $O^{5}$ refers to the final sentimental feature $E\;=\;O^{5}$, “without” means that E is the concatenation of $O^{4},O^{3},X^{5}$,without $O^{5}$. The bold score is the highest score of each column.

Feature	Only	Without
$O^{l}=Z^{l}\times M^{l}$	90.004	-
global	-	89.789
$O^{5}$	89.011	88.393
$O^{4}$	88.668	88.484
$O^{3}$	81.250	89.675
**ours**	**90.31**	**90.31**

**Table 3 sensors-21-02136-t003:** Classification results of different state-of-the-art methods on five different datasets: FI, ArtPhoto, Abstract, IAPSsubset, EmotionROI. The bold score is the highest score of each column.

Method	FI	ArtPhoto	Abstract	IAPSsubset	EmotionROI
LCH [38]	45.37	64.33	70.93	52.84	63.79
GCH [38]	47.95	66.53	67.33	69.96	66.85
PAEF [39]	58.42	68.42	66.23	65.77	73.45
Rao(a) [40]	62.79	71.53	67.82	78.34	74.51
SentiBank [16]	56.47	67.33	64.30	80.57	65.73
AlexNet [41]	68.63	69.27	65.49	84.58	71.60
VGG-16 [42]	73.95	70.48	65.88	87.20	72.49
ResNet101 [37]	75.76	71.08	66.64	88.15	73.92
DeepSentiBank [17]	64.39	70.26	69.07	86.31	70.38
PCNN [32]	73.59	71.47	70.26	88.65	74.06
Rao(b) [9]	79.54	74.83	71.96	90.53	78.99
Zhu [43]	84.26	75.50	73.88	91.38	80.52
AR [28]	86.35	74.80	76.03	92.39	81.26
Rao(c) [4]	87.51	78.36	77.28	93.66	82.94
**ours**	**90.31**	**79.24**	**78.14**	**94.82**	**85.10**

**Table 4 sensors-21-02136-t004:** Classification results of different state-of-the-art methods on twitter I and twitter II. The bold score is the highest score of each column.

Method	Twitter I			Twitter II
	Twitter I_5	Twitter I_4	Twitter I_3	
GCH [38]	67.91	67.20	65.41	77.68
LCH [38]	70.18	68.54	65.93	75.98
PAEF [39]	72.90	69.61	67.92	77.51
SentiBank [16]	71.32	68.28	66.63	65.93
DeepSentiBank [17]	76.35	70.15	71.25	70.23
PCNN [32]	82.54	76.52	76.36	77.68
VGG-16 [42]	83.44	78.67	75.49	71.79
AR [28]	88.65	85.10	81.06	80.48
RA-DLNet [26]	89.10	83.20	81.30	**81.20**
GM_EI_ &LRM_SI_ [27]	89.50	**86.97**	81.65	80.97
**ours**	**89.77**	**86.57**	**83.88**	**81.19**

## Data Availability

Not applicable.

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
