# Peer review of "Multi-Level Context Pyramid Network for Visual Sentiment Analysis"

_sensors, 2021, doi:10.3390/s21062136_

Round 1
Reviewer 1 Report
The paper addresses the problem of visual sentiment analysis. The author proposed a novel approach which implements a multi-scale region analysis by means of a pyramid architecture able to correlate sentiments with localized small semantic objects at different scales.
In particular, the architecture is based on ResNet101, plus some MACM modules which perform the multiscale analysis. Then, the output are combined and the classification is performed.
The approach allows also to visualize the regions associated with the classification.
The method is well detailed in all its parts, indeed the approach is clear and well presented. The experiments have been conducted on an high number and variety of datasets, and comparisions have been performed with several methods. Moreover, the authors presented an ablation study to demonstrate the contribute of each model component.
However, there are important missing works in the bibliography, I strongly suggest to read the following paper, which also contains the most important publications in the field, organized and presented based on several points of view:
Ortis, Alessandro, Giovanni Maria Farinella, and Sebastiano Battiato. "Survey on visual sentiment analysis." IET Image Processing 14.8 (2020): 1440-1456.
Reviewer 2 Report
This paper proposes a multi-level context pyramid network (MCPNet) for visual sentiment analyses. The proposed network is able to combine both local and global representations for sentiment prediction. The authors performed extensive experiments on seven benchmark datasets to demonstrate the superiority of the proposed model. The presentation is clear and the whole paper reads smooth. I think the authors can further improve their paper through the following aspects:
- I appreciate the authors' great efforts in presenting their methodology. However, I think the authors can put more emphasis on highlighting the novelty of the proposed method. For example, what is the difference between authors' "multi-scale" and other multi-scale CNN-based methods. Why is the authors' method specially effective for sentiment representation?
2. The authors compared their method with a few other methods. Have the authors tried to compare with other sentiment-focused methods, such as [1-3]?
[1] Campos, V., Jou, B., & Giro-i-Nieto, X. (2017). From pixels to sentiment: Fine-tuning CNNs for visual sentiment prediction. Image and Vision Computing, 65, 15-22.
[2] Fan, S., Jiang, M., Shen, Z., Koenig, B. L., Kankanhalli, M. S., & Zhao, Q. (2017, October). The role of visual attention in sentiment prediction. In Proceedings of the 25th ACM international conference on Multimedia (pp. 217-225).
[3] Yadav, A., & Vishwakarma, D. K. (2020). A deep learning architecture of RA-DLNet for visual sentiment analysis. Multimedia Systems, 26, 431-451.
3. The analyses of the experiments are nice. The authors show vivid visualizations on two datasets. However I think it can be further improved, so that the paper can provide more insights to fellow researchers. For example, among the 7 benchmark dataset, why is the proposed method most advantageous on the FI dataset? How different number of scales correlate to different dataset, or even different image content?
4. Minor points: the font size in Fig 3 is too small. When the authors mention the datasets such as COCO, it is better to give citations.
Reviewer 3 Report
The paper faces a challenging problem with a certain interest in the computer vision community.
In general, it is well written and sufficiently clear, motivations of the choices done can be often understood, methodology is almost always correctly presented, results are sufficiently convincing. I believe therefore that the paper could be accepted for publication.
However, some revision is necessary as suggested in the following.
Section 1 and Section 2
Some interesting or relevant recent works on visual sentiment analysis are not cited.
Just as an example:
-Lifang Wu et al., Visual Sentiment Analysis by Combining Global and Local Information, Neural Processing Letters, 51, 2063-2075, 2020.
-A. Ortis et al., Survey on visual sentiment analysis, IET Image Processing, 14, 8, 1440-1456, 2020.
- S. Surekha, Deep Neural Network-based human emotion recognition by computer vision, Advances in Electrical and Computer Technologies, 453-463, 672, 2020, Springer LNEE.
I suggest the authors to revise Section 2, and partially also Section 1, not only including the presentation of pertinent works not mentioned but also reorganizing the text appropriately.
Section 3
- Caption of Fig. 2 should be completed, concat, GAP and FC are not defined. Also complete caption of Fig. 3.
- ResNet101 is used as backbone of the approach developed. I suggest to include a short paragraph providing the key features of the tool together with a stronger motivation about its choice
- please take care of the correct definition of all the items cited in the different contexts (e.g., c, O, M, X, scale, …, FPN, …) and be coherent in the use of variables (e.g., c, l).
- in general, when some change is proposed w.r.t. a previous developed method, the authors should explain in detail why (e.g., up-sampling operation, line 270)
- sub-section 3.4 is very specific and its inclusion here is not well motivated. I suggest to move any comment to sub-section 4.5, accordingly reorganizing the presentation.
Section 4
- Lines 328-336, the explanation given for the choice done requires a better discussion. Here it is only introduced but strong motivations are lacking. Since this decision affected the subsequent work, it would be appropriate to provide adequate explanations.
- Lines 345v - 354, I imagine that all the thresholds mentioned are the result of some optimization process but it would be good to give more precise explanations.
- captions of Figs. 10, 11 and 13 should be improved.
One final point: many phases of the developed method are taken up or modify pre-existing approaches. In my opinion, it is a little simplistic to mention only the papers considered, but the authors should not only highlight the specific aspects of these works but also give prominence to the changes made, so as to fully appreciate the originality of their contribution.
Round 2
Reviewer 2 Report
The authors addressed most of my concerns in the reponse. The paper has been largely improved by adding more description of the proposed architecture, as well as more comparisons of the state-of-the-art methods. I only have a few minor comments: 1) The authors give the citation for ResNet101 later in the paper. The citation should be given when ResNet101 is mentioned for the first time in the main text. 2) Fig 3 caption, please do not start with the word "And" in a formal research paper. 3) The English launage can be furhter improved. For example, in the abstract, the phrase "especially the accuracy on the FI dataset exceeds 90%" is unclear and seems to be grammaticall inaccurate.